# *GSM1* Requires Hap4 for Expression and Plays a Role in Gluconeogenesis and Utilization of Nonfermentable Carbon Sources

**DOI:** 10.3390/genes15091128

**Published:** 2024-08-27

**Authors:** Manika Bhondeley, Zhengchang Liu

**Affiliations:** 1Department of Biological Sciences, University of New Orleans, New Orleans, LA 70148, USA; 2Kudo Biotechnology, 117 Kendrick Street, Needham, MA 02494, USA

**Keywords:** transcriptional regulation, Gsm1, Hap2/3/4/5, Hap4, *S. cerevisiae*, Cat8, Fbp1, Pck1, gluconeogenesis

## Abstract

Multiple transcription factors in the budding yeast *Saccharomyces cerevisiae* are required for the switch from fermentative growth to respiratory growth. The Hap2/3/4/5 complex is a transcriptional activator that binds to CCAAT sequence elements in the promoters of many genes involved in the tricarboxylic acid cycle and oxidative phosphorylation and activates gene expression. Adr1 and Cat8 are required to activate the expression of genes involved in the glyoxylate cycle, gluconeogenesis, and utilization of nonfermentable carbon sources. Here, we characterize the regulation and function of the zinc cluster transcription factor Gsm1 using Western blotting and *lacZ* reporter-gene analysis. *GSM1* is subject to glucose repression, and it requires a CCAAT sequence element for Hap2/3/4/5-dependent expression under glucose-derepression conditions. Genome-wide CHIP analyses revealed many potential targets. We analyzed 29 of them and found that *FBP1*, *LPX1*, *PCK1*, *SFC1*, and *YAT1* require both Gsm1 and Hap4 for optimal expression. *FBP1*, *PCK1*, *SFC1*, and *YAT1* play important roles in gluconeogenesis and utilization of two-carbon compounds, and they are known to be regulated by Cat8. *GSM1* overexpression in *cat8*Δ mutant cells increases the expression of these target genes and suppresses growth defects in *cat8*Δ mutants on lactate medium. We propose that Gsm1 and Cat8 have shared functions in gluconeogenesis and utilization of nonfermentable carbon sources and that Cat8 is the primary regulator.

## 1. Introduction

The budding yeast *S. cerevisiae* can utilize different types of carbon sources to produce energy. In the presence of abundant glucose, yeast cells prefer to use fermentation to produce ATP, even in the presence of oxygen and non-fermentable carbon sources. During the diauxic shift or when the glucose level is low, there is a switch to respiratory metabolism for the generation of ATP, which requires a coordinated change in gene expression via multiple transcriptional regulatory factors, as reviewed in [1,2]. This leads to increased expression of genes encoding enzymes and proteins involved in gluconeogenesis, the glyoxylate cycle, the tricarboxylic acid cycle, and oxidative phosphorylation.

The change in metabolic programming during the transition to glucose-limiting conditions is regulated by a variety of regulatory proteins, including the Hap2/3/4/5 complex, Snf1, Adr1, Cat8, Rds2, Ert1, and Sip4 [3,4,5,6,7,8,9,10,11,12]. Snf1 is a subunit of a heterotrimeric complex that is activated upon phosphorylation by upstream kinases and promotes the glucose-derepression pathway for the utilization of alternative carbon sources by regulating the phosphorylation state of Cat8, Sip4, and Adr1 [5,6,8,13,14,15]. Snf1 is also involved in transcriptional control of Cat8 [5]. Adr1 is a zinc finger transcription factor that regulates genes for the utilization of lactate, glycerol, and ethanol [3,4]. Cat8, Sip4, Rds2, and Ert1 are zinc cluster transcription factors. Cat8 and Sip4 are subject to glucose repression, and *SIP4* expression requires Cat8 [6,7,15]. Cat8 and Sip4 share common target genes, including those involved in the glyoxylate cycle and gluconeogenesis [6,16]. The targets of Rds2 include genes involved in gluconeogenesis, the tricarboxylic acid cycle, and the glyoxylate cycle [11]. The target genes of Ert1, as identified by chromatin immunoprecipitation assays, overlap with those of Adr1, Cat8, and Rds2 [10]. Genome-wide location analysis and transcriptome analysis reveal important overlaps among the targets of the transcriptional regulators Adr1, Cat8, Ert1, and Rds2 [4,11,12,17], indicating that yeast utilization of non-fermentable carbon sources requires an intricately regulated network of factors and their target genes.

Utilization of non-fermentable carbon sources requires ATP production via oxidative phosphorylation in the mitochondria. The Hap2/3/4/5 complex is a master regulator of mitochondrial biogenesis and energy metabolism in yeast [18,19,20]. Hap2/3/4/5 is a heterotetrameric complex that is activated by heme and regulates the transcription of genes involved in the tricarboxylic acid cycle and oxidative phosphorylation [21,22,23,24,25,26,27]. Hap4 is the regulatory subunit of this complex, and its expression is subject to glucose repression. The Hap2/3/5 trimer binds to CCAAT sequence elements in target gene promoters and requires Hap4 for transcriptional activation [28].

Gsm1, a zinc cluster transcription factor, has been proposed to regulate the expression of genes involved in gluconeogenesis (*PCK1* and *FBP1*, W.G. Bao & M. Bolotin-Fukuhara, personal communication to the authors of the review paper) [1]. Genome-wide transcriptome analysis has shown that the expression of *GSM1/YJL103C* is similar to that of genes involved in cellular respiratory metabolism and that *GSM1/YJL103C* requires Hap2 for expression under glucose-derepression conditions [18,29]. Several chromatin immunoprecipitation (CHIP) analyses have identified potential Gsm1 targets [30,31,32], but none has been validated using gene or protein expression assays. A recent report shows that Rds2 and Gsm1 have overlapping and distinct targets [30]. The authors reported that *gsm1*Δ had no growth defects on non-fermentable carbon sources by itself or in combination with mutations in other genes involved in gluconeogenesis. In this report, we characterized the regulation of *GSM1* and dissected its functions. We found Hap4 is essential for *GSM1* expression under glucose-derepression conditions. We validated a number of Gsm1 targets and uncovered combinatorial regulation of gene expression by Gsm1 and Cat8. We demonstrate that Gsm1 has functions overlapping with those of Cat8 and that it is required for cell growth on lactate medium in *cat8*Δ mutant cells.

## 2. Materials and Methods

### 2.1. Growth Media, Growth Conditions, Strains, and Plasmids 

Yeast strains were grown at 30 °C in YPD (1% Bacto Yeast Extract (Fisher Scientific, Waltham, MA, USA), 2% Bacto Peptone (Fisher Scientific), 2% glucose (Fisher Scientific)), YPL (1% Bacto Yeast Extract, 2% Bacto Peptone, 3.7% DL-Lactic acid (85%) (Sigma-Aldrich, St. Louis, MO, USA), adjusted to pH 5.3 using NaOH), YNBcasD (0.67% yeast nitrogen base (Fisher Scientific), 1% casamino acids (Fisher Scientific), 2% dextrose), YNBcas5D (similar to YNBcasD, with 5% dextrose), YNBcasR (0.67% yeast nitrogen base, 1% casamino acids, 2% raffinose (USBiological, Salem, MA, USA)), and a complete supplement mixture medium with raffinose as the carbon source (CSM-Raffinose) (0.67% yeast nitrogen base, 2% glucose, 0.6 g/L CSM minus uracil and leucine, 2% raffinose), as indicated in the text or in the figure legends. Amino acids and uracil were added to the growth medium at standard concentrations to cover auxotrophic requirements if required [33]. Agar (USBiological) was added at a final concentration of 2% for the solid medium. Cells were grown in liquid medium in a shaking incubator at 220 rpm and at 30 °C. Cells streaked on plate medium were incubated at 30 °C. The yeast strains and plasmids used in this study are listed in Table 1 and Table 2. Deletion mutant strains were constructed by transforming yeast with the required gene-knockout cassettes, dissecting sporulated heterozygous diploid strains, and/or crossing two mutant strains to obtain diploids for sporulation and dissection. Gene-deletion mutations were confirmed by PCR genotyping.

### 2.2. Yeast Transformation and β-Galactosidase Activity Assays

Yeast cells were freshly grown in YPD liquid medium and transformed using the high-efficiency method [38]. The YNBcasD medium, the SD medium supplemented with appropriate amino acids and uracil, and the YPD medium supplemented with geneticin were used to select yeast transformants based on the *URA3*, *HIS3*, and *kanMX4* selection markers, respectively. For the β-galactosidase activity assays, the yeast strains were grown in growth medium as indicated in the text or in the figure legend at 30 °C for at least six generations to allow them to reach an OD_600_ of about 0.6 before collection. The cells were collected by centrifugation, and the β-galactosidase activity assays were conducted using o-nitrophenyl β-D-galactopyranoside (ONPG) as substrate, as described [33]. Two to six independent cultures were grown, and assays were carried out in duplicate for each sample. The data are presented as the mean ± standard deviation. The means of the β-galactosidase activity assay results were compared using a *t*-test. A “∗” in figures indicates a significant difference between the means of two groups of data (*p* < 0.05).

### 2.3. Serial Dilution of Cells for Growth Analysis

The wild-type and isogenic mutant strains were freshly grown on the YPD solid medium at 30 °C for 2–3 days. The cells were picked from a plate into sterile water and diluted to the same starting OD_600_ of 0.1. Five-fold serial dilutions were made using sterile 96-well plates and 8-channel pipettes. Then, 5 μL aliquots of serially diluted cell resuspensions were spotted on solid YPD and YPL solid media. The cells were grown for 2 to 4 days at 30 °C before pictures were taken for cell-growth analysis.

### 2.4. Cellular Extract Preparation, Immunoblotting, and Immunoprecipitation

The yeast strains were grown in growth medium as indicated at 30 °C for at least six generations to allow them to reach an OD_600_ of about 0.6, and total cellular proteins were prepared as described [39]. Briefly, 1 mL cell culture was mixed with 160 μL freshly prepared solution of 7.5% β-mercaptoethanol and 1.85N NaOH and incubated on ice for 10 min. Then, 84 μL 100% trichloroacetic acid (*w*/*v*) was then added, and the mixture was incubated on ice for 10 min before protein pellets were obtained by centrifugation at 21,000 g for 5 min. Protein samples were resuspended in SDS-PAGE sample buffer with 100 mM dithiothreitol and boiled for 3 min before being separated by SDS-polyacrylamide gel (7.5%) electrophoresis. The lanes were loaded with equivalent amounts of proteins based on the OD_600_ reading of the cell cultures. Pre-stained protein ladder (P7710S, New England Biolabs, Ipswich, MA, USA) was used in all protein gels. Proteins were transferred to the nitrocellulose membrane for immunoblotting. The following antibodies were used in this study: anti-GFP antibody B-2 (1:1000, Santa Cruz Biotechnology Inc., Dallas, TX, USA); anti-Pgk1 antibody (1:2000), rabbit polyclonal antibodies against recombinant yeast phosphoglycerate kinase generated by the lab; and HRP-conjugated goat anti-mouse secondary antibody (1:3000, catalog # 115-035-003, Jackson ImmunoResearch Laboratories, West Grove, PA, USA). Chemiluminescence images of Western blots were captured using the Bio-Rad ChemiDoc MP imaging system (Hercules, CA, USA) and processed using Bio-Rad Image Lab software (version 6.1).

## 3. Results

### 3.1. GSM1 Expression Is Subject to Glucose Repression and Requires Hap4 under Glucose-Derepression Conditions

Large-scale gene expression profiling studies have shown that *GSM1* expression increases during the diauxic shift and during growth on glycerol or ethanol [29,40]. To confirm that *GSM1* expression is carbon source-dependent, we generated a *GSM1−lacZ* reporter gene by fusing a 924-bp promoter of *GSM1* to the bacterial *lacZ* gene. The plasmid encoding the *GSM1−lacZ* reporter gene was transformed into a wild-type BY4741 strain, and transformants were grown in media with dextrose or raffinose as the sole carbon source. Raffinose is a trisaccharide and a glucose-derepression carbon source. In the wild-type strain, a β-galactosidase activity assay shows that the expression of *GSM1-lacZ* is very low in glucose-grown cells, at a level close to the detection limit of the β-galactosidase activity assay using ONPG as the substrate. There is 5.2-fold higher expression in raffinose-grown cells compared to dextrose-grown cells (Figure 1A), a finding consistent with published microarray data.

Many genes require the Hap2/3/4/5 complex for their expression in cells grown under glucose-derepression conditions [18]. In the same study, a *hap2*∆ mutation was reported to reduce the expression of *GSM1*. To determine whether *GSM1* is a target of Hap4, we introduced the plasmid encoding the *GSM1-lacZ* reporter gene into *hap4*∆ mutant cells and β-galactosidase activity assays were carried out. Figure 1A shows that a *hap4*∆ mutation reduces *GSM1-lacZ* expression by 45% in glucose-grown cells and by 7.3-fold in raffinose-grown cells, indicating that *GSM1* expression is under the control of Hap4. In wild-type cells, there was a 5.2-fold increase in *GSM1-lacZ* activity in raffinose-grown cells compared to glucose-grown cells (Figure 1A). This increase in *GSM1* expression was completely blocked in *hap4*∆ mutant cells. Together, our data suggest that *GSM1* is a new target of Hap4.

Our data in Figure 1A clearly indicates that *hap4*Δ reduces *GSM1* expression in cells grown in raffinose medium. A change in gene transcription does not always lead to a corresponding change in the protein level. Therefore, to determine whether the transcriptional control of *GSM1* correlates with the protein level of Gsm1, we generated a plasmid encoding Gsm1 with a C-terminal GFP tag and introduced it into *gsm1*∆ cells. Transformants were grown in dextrose and raffinose medium. As a control, *gsm1*∆ cells carrying an empty vector were cultured similarly. The expression of the Gsm1-GFP fusion protein was detected by Western blotting with an anti-GFP antibody. *gsm1*∆ cells carrying the plasmid encoding the Gsm1-GFP fusion protein cultured in raffinose medium, but not those cultured in glucose medium, yielded a single band of ~100 kD, close to the predicted size of 98.7 kD (Figure 1B). Similar results were also obtained from an otherwise wild-type strain expressing *GSM1-GFP* (Figure 1C). In contrast, *hap4*Δ mutant cells expressing *GSM1-GFP* grown in raffinose medium did not yield a visible signal for Gsm1-GFP. Together, our data indicate that expression of *GSM1* under glucose-derepression conditions requires Hap4.

### 3.2. A Proximal CCAAT Sequence Element Is Required for GSM1 Expression

The Hap2/3/4/5 complex activates the expression of its target genes by binding to CCAAT sequence elements in their promoter region under glucose-derepression conditions, as reviewed in [41]. The promoter region of *GSM1* contains two CCAAT sequence elements at −634 bp and at −286 bp in relation to the ATG start codon (Figure 2A). To determine which CCAAT element confers Hap4-dependent *GSM1* expression, we mutated each of the CCAAT sequences into the TCACA sequence and generated a site 1 mutation (S1M) at the −634 bp position and a site 2 mutation (S2M) at the −286 bp position (Figure 2A). The effect of the two mutations in the reporter constructs *GSM1(S1M)-lacZ* and *GSM1(S2M)-lacZ* was examined in wild-type BY4741 cells grown in dextrose and raffinose media. In both dextrose- and raffinose-grown cells, the CCAAT-element mutation at the −634 bp position (*GSM1(S1M)-lacZ*) did not lead to significant changes in the expression of the *lacZ* reporter gene. In contrast, the mutation in the CCAAT sequence element at the −286 bp position reduced *GSM1-lacZ* expression by 39% in glucose-grown cells and by 7.4-fold in raffinose-grown cells. The almost identical effects on *GSM1-lacZ* expression caused by a *hap4*∆ mutation and a mutation in the proximal CCAAT sequence element (compare Figure 1A with Figure 2B) indicate that Hap4 directly regulates the expression of *GSM1* via the CCAAT sequence at the −286 bp position.

### 3.3. Identification of Potential Gsm1 Target Genes via Analysis of lacZ Reporter Genes

A genome-wide study of Gsm1 binding sites revealed potential Gsm1 target genes via a chromatin immunoprecipitation (CHIP) analysis [31]. The authors introduced a new method that involved two rounds of T7 RNA polymerase amplification (double-T7) to amplify ChIP DNA for microarray analysis. The double-T7 method showed stronger binding signals compared to traditional ligation-mediated polymerase chain reaction (LM-PCR). Figure 3A shows an example, *FBP1*, as a potential target of Gsm1 that was identified using these two methods. We used the data from this study to select target genes and chose 29 potential targets (*ADR1, ERG3, FBP1, GAT2, GID7, GID8, HAP4, IDP2, LPX1, MDH2, MDY2, MIR1, MOT3, MSN2, MSN4, PCK1, PRB1, PYC1, PYK2, RCF2, RHO5, ROX1, SFC1, SHR5, SIT4, TSL1, VHR1, YAT1*, and *ZWF1*) based on the binding signals from both the T7-based amplification method and conventional LM-PCR. Two of the 29 genes, *ADR1* and *HAP4*, were also identified in another Gsm1 CHIP assay [32]. While we were preparing the manuscript, another report on the Gsm1 targets was published; this report included *PCK1*, *HAP4*, and *FBP1* as potential targets [30]. In all three reports on Gsm1 targets identified via CHIP analysis, the target genes were not confirmed by gene- or protein-expression analysis. We generated 29 *lacZ* reporter genes, each of which was under the control of a different gene promoter listed above. Plasmids bearing *lacZ* reporter genes were transformed into wild-type and *gsm1*∆ cells. Transformants were grown in raffinose medium and β-galactosidase activity assays were conducted. Figure 3B shows that the expression of 10 reporter genes, namely, *FBP1, IDP2, LPX1, MDY2, MSN4, PYC1, PYK2, YAT1, ZWF1,* and *GAT2*, was significantly reduced (*p* < 0.05) in *gsm1*∆ mutant cells. Five out of these 29 genes, *GID7*, *MDH2*, *PCK1*, *SFC1*, and *SHR5*, show reduced expression, with a *p* value close to the 0.05 cut-off, in *gsm1*∆ mutant cells compared to wild-type cells. Most of the *lacZ* reporter genes show reduced expression in *gsm1*∆ mutant cells, a finding consistent with the notion that Gsm1 is a transcriptional activator. Only two genes, *VHR1* and *HAP4*, show an increase in expression of more than 5% in *gsm1*∆ cells.

Among the 15 genes showing a reduction in their expression in *gsm1*∆ cells with a *p* value less than or close to 0.05, nine of them, *FBP1*, *IDP2*, *LPX1*, *PCK1*, *PYC1*, *PYK2*, *SFC1*, *YAT1*, and *ZWF1*, are involved in gluconeogenesis, carbohydrate metabolism, and/or the utilization of non-fermentable carbon sources. We chose five for further analysis: *FBP1* (encoding fructose-1,6-bisphosphatase), *PCK1* (encoding phosphoenolpyruvate carboxykinase), *SFC1* (encoding a mitochondrial succinate-fumarate transporter), *LPX1* (encoding a peroxisomal matrix-localized lipase), and *YAT1* (encoding outer mitochondrial carnitine acetyltransferase). Figure 3C shows that expression of *FBP1-*, *PCK1-*, *LPX1-*, *SFC1-*, and *YAT1-lacZ* reporter gene are reduced in both *gsm1*Δ and *hap4*Δ mutant cells grown in the raffinose medium. Since Hap4 is strictly required for *GSM1* expression in this medium (Figure 1), it is expected that *hap4*Δ reduces the expression of these five genes. In fact, compared to *gsm1*Δ, *hap4*Δ leads to a greater reduction in the expression of *FBP1*, *LPX1*, *SFC1*, and *YAT1*, suggesting that Hap4 may regulate other transcription factors, which in turn mediate the expression of these four genes.

We have shown that Hap4 is required for *GSM1* expression under glucose-derepression conditions (Figure 1). The three reports on the CHIP analysis of Gsm1 all show *HAP4* is a potential target [30,31,32]. However, we failed to see reduced expression of *HAP4-lacZ* in *gsm1*Δ mutant cells (Figure 3C). On the contrary, among the 29 genes we analyzed, *HAP4-lacZ* was the only reporter gene that showed a small induction, with a *p* value close to 0.05 (*p* = 0.16), in *gsm1*Δ cells (Figure 3B). In contrast, *hap4*Δ reduces *HAP4* expression by 2.3-fold, a result consistent with published findings [27]. It appears that a strong binding based on CHIP data does not necessarily translate into transcriptional regulation. Rds2 has been reported to be required for *HAP4* expression [11], but we failed to see decreased expression of *HAP4-lacZ* in *rds2*Δ mutant cells (BY4741 background) grown in raffinose. The discrepancy may be attributed to a difference in the growth conditions (ethanol versus raffinose).

### 3.4. GSM1 Overexpression Increases Expression of FBP1-, LPX1-, PCK1-, SFC1-, and YAT1-lacZ Reporter Genes in hap4∆ Mutant Cells

We have shown that Hap4 is strictly required for *GSM1* expression in cells grown in the raffinose medium. Figure 3C shows that the effect of *hap4*Δ on reducing the expression of the *FBP1-, LPX1-*, *PCK1-, SFC1-,* and *YAT1-lacZ* reporter genes is stronger than that of *gsm1*Δ. We decided to test the effect of *GSM1* overexpression on the expression of these five reporter genes in both wild-type and *hap4*Δ mutant cells. Accordingly, wild-type and *hap4*Δ mutant cells expressing the *lacZ* reporter genes were transformed with a plasmid encoding Gsm1 with a C-terminal 3× myc tag under the control of the strong *TEF2* promoter (*TEF2-GSM1-myc*) or with the empty vector pRS415TEF, which served as a control. Transformants were selected on complete supplement mixture medium without uracil and leucine to select for and maintain the plasmid carrying the *URA3* or *LEU2* marker. Transformants were grown in CSM-Raffinose medium without uracil and leucine, and a β-galactosidase activity assay was conducted. Figure 4A shows that *TEF2-GSM1-myc* increases the expression of these five *lacZ* reporter genes in both wild-type and *hap4*Δ mutant cells, consistent with the notion that these five genes are Gsm1 targets. In all reporter genes except *YAT1-lacZ*, *TEF2-GSM1-myc* leads to a significantly higher expression level in wild-type than in *hap4*Δ mutant cells, suggesting that other Hap4-dependent proteins also contribute to the regulation of *FBP1*, *LPX1*, *PCK1*, and *SFC1*.

After establishing a functional essay for *GSM1*, we decided to determine whether the *GSM1-GFP* construct used to produce the results in Figure 1B,C was functional. *gsm1*Δ mutant cells expressing *FBP1-lacZ* were transformed with a plasmid encoding *GSM1-GFP* under the control of the *GSM1* promoter or with an empty vector, which served as the control. Transformants were grown in CSM-raffinose medium, and a β-galactosidase activity assay was conducted. Figure 4B shows that *GSM1-GFP* largely complements *gsm1*Δ by restoring *FBP1-lacZ* expression. Since Gsm1-GFP was largely functional, we can conclude that the protein-expression result presented in Figure 1B,C is physiologically relevant.

### 3.5. Cat8 and Gsm1 Are Important in the Transcriptional Regulation of FBP1, PCK1, SFC1, and YAT1 and in the Utilization of Lactate

Yeast cells undergo a transcriptional switch in order to utilize nonfermentable carbon sources. There is a dramatic upregulation of genes involved in gluconeogenesis, the glyoxylate cycle, the tricarboxylic acid cycle, and oxidative phosphorylation. Genome-wide transcriptome analyses have shown that Cat8 and Adr1 are required for derepression of many genes important for utilization of nonfermentable carbon sources [4,16,17]. Among the five genes we chose for further analysis as Gsm1 targets, *FBP1*, *PCK1*, *SFC1*, and *YAT1* are known to be regulated by Cat8 [16]. To analyze whether *FBP1*, *LPX1*, *PCK1*, *SFC1*, and *YAT1* are subject to combinatorial regulation by Gsm1, Adr1, and Cat8, we generated all possible double and triple mutants from *cat8*∆, *gsm1*∆, and *adr1*∆ mutations. We transformed *FBP1-lacZ*, *LPX1-lacZ*, *PCK1-lacZ*, *SFC1-lacZ*, and *YAT1-lacZ* reporter constructs into *adr1*∆, *cat8*∆, *gsm1*∆, *adr1*∆ *cat8*∆, *adr1*∆ *gsm1*∆, *cat8*∆ *gsm1*∆, and *adr1*∆ *cat8*∆ *gsm1*∆ mutant cells. Transformants were grown in YNBcasR medium, and a β-galactosidase activity assay was carried out. Figure 5A,C,E shows that there is combinatorial regulation of the *FBP1*-, *PCK1*-, and *YAT1*-lacZ reporter genes by Cat8 and Gsm1, since the expression levels of these three *lacZ* reporter genes in the *cat8*∆ *gsm1*∆ double mutant are lower than those observed in the *cat8*∆ and *gsm1*∆ single mutants. Although *SFC1-lacZ* is significantly reduced in *gsm1*∆ mutant cells, it is difficult to determine whether *SFC1* is subject to combinatorial regulation by Cat8 and Gsm1 due to the extremely low expression level already present in the *cat8*Δ single mutant (Figure 5D).

An *adr1*Δ mutation does not result in significant changes in the expression of the *FBP1*-, *PCK1*-, *SFC1*-, and *YAT1*-lacZ reporter genes (Figure 5A,C–E), which is consistent with published findings [4]. Consistently, while Cat8 and Gsm1 are required for the expression of these four reporter genes, any additional combinatorial regulation by Adr1 is either minimal or nonexistent. Together, our data suggest that Cat8 and Gsm1 have related functions and that Cat8 is the primary regulator of the two.

Adr1 and Cat8 play important roles in the regulation of genes involved in the utilization of nonfermentable carbon sources. Our finding that Cat8 and Gsm1 have related functions prompted us to test the growth phenotypes of *adr1*∆, *cat8*∆, *gsm1*∆, *adr1*∆ *cat8*∆, *adr1*∆ *gsm1*∆, *cat8*∆ *gsm1*∆, and *adr1*∆ *cat8*∆ *gsm1*∆ mutant cells on growth medium with dextrose or lactate as the sole carbon source (Figure 5F). On the dextrose medium, none of the mutants exhibited growth defects, which is consistent with the role of these transcription factors in the utilization of nonfermentable carbon sources. On the lactate medium, *cat8*Δ led to significant growth defects while *adr1*∆, *gsm1*∆, *adr1*∆ *gsm1*∆ mutant cells had the same level of growth as wild-type cells. Importantly, while *gsm1*∆ did not lead to growth defects, the residual growth in *cat8Δ* mutant cells was eliminated in the *cat8*∆ *gsm1*∆ double- and *adr1*∆ *cat8*∆ *gsm1*∆ triple mutant cells. Together, our data suggest that Cat8 and Gsm1 have related functions and that Cat8 is the primary regulator.

### 3.6. GSM1 Overexpression Increases Target-Gene Expression in adr1∆ cat8∆ Mutant Cells and Suppresses Growth Defects Associated with cat8Δ

Our data, as represented in Figure 5, show that Cat8 and Gsm1 have related functions in the regulation of genes involved in the utilization of nonfermentable carbon sources. Adr1 is also important for this function. We decided to test whether *GSM1* overexpression can increase the expression of the target genes in the absence of both Adr1 and Cat8. Accordingly, wild-type and *adr1*∆ *cat8*∆ mutant cells expressing *FBP1-lacZ, LPX1-lacZ, PCK1-lacZ, SFC1-lacZ,* and *YAT1-lacZ* reporter genes were transformed with a plasmid encoding *GSM1* with a 3x myc tag at the C-terminus under the control of the strong *TEF2* promoter. Transformants were grown in CSM-raffinose medium, and a β-galactosidase assay was conducted. Figure 6A shows that *GSM1* overexpression in *adr1*∆ *cat8*∆ mutant cells significantly increased the expression of all five reporter genes, indicating that Gsm1 can activate the expression of its target genes independent of Adr1 and Cat8. Figure 6A also shows that *GSM1* overexpression in wild-type cells led to higher levels of target gene expression than did its overexpression in *adr1*∆ *cat8*∆ mutant cells. This result suggests that, although Gsm1 can function independently of Adr1 and Cat8, it requires Adr1 and/or Cat8 for maximum expression of *FBP1-lacZ, LPX1-lacZ, PCK1-lacZ, SFC1-lacZ,* and *YAT1-lacZ* reporter genes.

Next, we wanted to determine whether *GSM1* overexpression can suppress the severe growth defects of *cat8*∆ mutant cells on lactate medium. We overexpressed *GSM1* under the control of the *TEF2* promoter in *adr1*∆, *cat8*∆, and *cat8*∆ *adr1*∆ cells, which were serially diluted and spotted on YPD and YPL media. Figure 6B shows that *GSM1* overexpression suppressed the growth defects of *cat8*∆ and *cat8*∆ *adr1*∆ double mutant cells on lactate medium. Together, our data indicate that Gsm1 and Cat8 have related functions via combinatorial regulation of target gene expression, which in turn enables cells to utilize non-fermentable carbon sources.

## 4. Discussion

In this report, we characterized the regulation and function of the zinc cluster transcription factor Gsm1. *GSM1* is subject to glucose repression and requires Hap4 for its induced expression under glucose-derepression conditions. A number of targets genes based in CHIP data were validated, including *FBP1, LPX1, PCK1, SFC1,* and *YAT1*. We found that *gsm1*Δ and *cat8*Δ have additive effects on the reduction of target gene expression and that *GSM1* overexpression suppresses the severe growth defects of *cat8Δ* mutant cells grown on a non-fermentable carbon source. We propose that Gsm1 and Cat8 have shared functions in regulating target gene expression and enabling yeast cells to utilize non-fermentable carbon sources.

*GSM1* expression is subject to glucose repression (Figure 1), a mechanism similar to that regulating the expression of *CAT8* and *SIP4*. This is achieved by Hap4-dependent activation under glucose-derepression conditions. The expression of *HAP4*, like that of its target genes, is also subject to glucose repression. In glucose-grown cells, low levels of *HAP4* expression translate to reduced activity of the Hap2/3/4/5 complex, which leads to basal expression of *GSM1*. The strategy of maintaining low expression levels of *GSM1*, *CAT8*, and *SIP4* is logical since they are not required in cells grown in glucose. The Hap2/3/4/5 complex binds to CCAAT sequence elements in the promoter of the target genes, leading to their transcriptional activation. The promoter of *GSM1* contains two CCAAT sequence elements. We found that the one close to the ATG start codon is essential for *GSM1* expression under glucose-derepression conditions, indicating that Hap2/3/4/5 regulation of *GSM1* expression is direct. Due to the interdependent regulation of genes encoding transcription factors involved in the transition to nonfermentable growth, it will be a challenge to determine whether the regulation of a target gene expression by a transcriptional regulator is direct or indirect.

The function of Gsm1 has not been clear until this study. Three genome-wide CHIP analyses have revealed potential Gsm1 targets, but they have not been validated by gene- or protein-expression assays. Among 29 genes we analyzed, we found that 12 genes, *FBP1, IDP2, LPX1, MDY2, MSN4, PCK1, PYC1, PYK2, SFC1, YAT1, ZWF1,* and *GAT2*, are Gsm1 targets. As shown in Figure 3B, the *p* values for the differences in the expression of *PCK1* and *SFC1* in wild-type cells versus in a *gsm1* mutant were 0.14 and 0.17, respectively. These values are not considered statistically different. However, the higher *p* values were due to the analysis of only two independent cultures of yeast strains expressing *PCK1* or *SFC1* while most strains carrying the reporter genes were represented by four independent cultures. As seen in Figure 4, Figure 5 and Figure 6, *PCK1* and *SFC1* are authentic targets of Gsm1. Among the 12 genes, *FBP1*, *PCK1,* and *PYC1* encode enzymes in the gluconeogenesis pathway and *SFC1* is important for the utilization of ethanol and acetate. Thus, Gsm1 is involved in gluconeogenesis and the utilization of nonfermentable carbon sources.

It has been proposed that Gsm1 is important for oxidative phosphorylation [40], a suggestion based on the analysis of genes having expression profiles similar to that of *GSM1*. Since *GSM1* is under the control of Hap2/3/4/5 complex, its expression profile should be similar to that of genes involved in the tricarboxylic acid cycle and oxidative phosphorylation. We believe this conclusion is misleading. *HAP4* has been highlighted as a potential *GSM1* target [30,31,32]. We failed to detect reduced *HAP4* expression in *gsm1*Δ mutant cells grown in raffinose medium (Figure 3C). On the contrary, a slight increase in the expression of *HAP4* in *gsm1*Δ mutant cells was detected. This may be an example in which the binding of a transcription factor to the promoter does not lead to altered gene expression. It is possible that Gsm1 binding can lead to increased expression of *HAP4* under other growth conditions.

Among the 12 genes we identified as Gsm1 targets, *FBP1, IDP2, PCK1, SFC1,* and *YAT1* are also targets of Cat8 [16]. In this report, we show that *cat8*Δ and *gsm1*Δ have additive effects on target gene expression and that *GSM1* overexpression can rescue reduced gene expression caused by *cat8*Δ (Figure 5 and Figure 6). We also found that *gsm1Δ* and *cat8Δ* have an additive effect in reducing cell growth on lactate medium and that *GSM1* overexpression can suppress the severe growth defects caused by *cat8*Δ (Figure 5 and Figure 6). Taking these results together, we propose that Gsm1 and Cat8 have related functions in gluconeogenesis and the utilization of non-fermentable carbon sources and that Cat8 is the primary regulator of the two. Identification of Gsm1 as another regulator of genes involved in the utilization of nonfermentable carbon sources adds to the already complex network of transcriptional regulators involved in the process. Future work is needed to determine why yeast cells employ so many regulators with overlapping functions and cross-regulations to utilize carbon sources.

## Figures and Tables

**Figure 1 genes-15-01128-f001:**
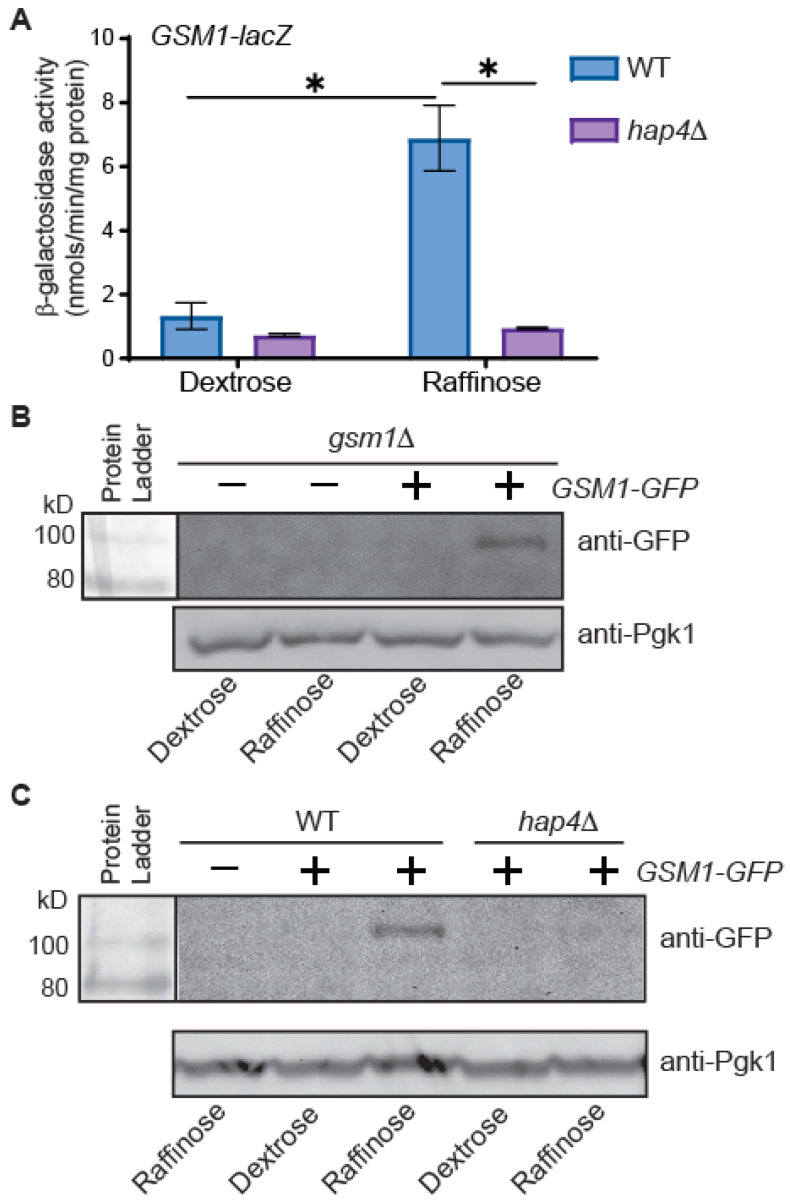
*GSM1* expression is subject to glucose repression and requires Hap4 for induced expression under glucose-derepression conditions. (**A**) Wild-type (BY4741) and *hap4*∆ mutant cells (ZLY2811) carrying a plasmid encoding the *GSM1-lacZ* reporter gene (pZL3454) were grown to mid-logarithmic phase in YNBcas5D (Dextrose) and YNBcasR (Raffinose) medium. β-galactosidase activity assays were conducted as described in the Materials and Methods. The data are presented as the mean ± standard deviation. The means of the results were compared by a *t*-test. “∗” indicates a significant difference (*p* < 0.05) between the means of two groups of data indicated by the beginning and end of the horizontal line. (**B**,**C**) Wild-type (BY4741), *hap4*∆ (ZLY2811), and *gsm1*∆ (MBY123) mutant cells carrying a plasmid encoding *GSM1-GFP* (pZL3462) as indicated were grown in YNBcas5D (dextrose) and YNBcasR (raffinose) medium to mid-logarithmic phase, and total cellular proteins were prepared and probed by Western blotting using an anti-GFP antibody, as described in the Materials and Methods. Pgk1 was included as a loading control. The result was representative of three independent experiments for panel (**B**) and of two independent experiments for panel (**C**).

**Figure 2 genes-15-01128-f002:**
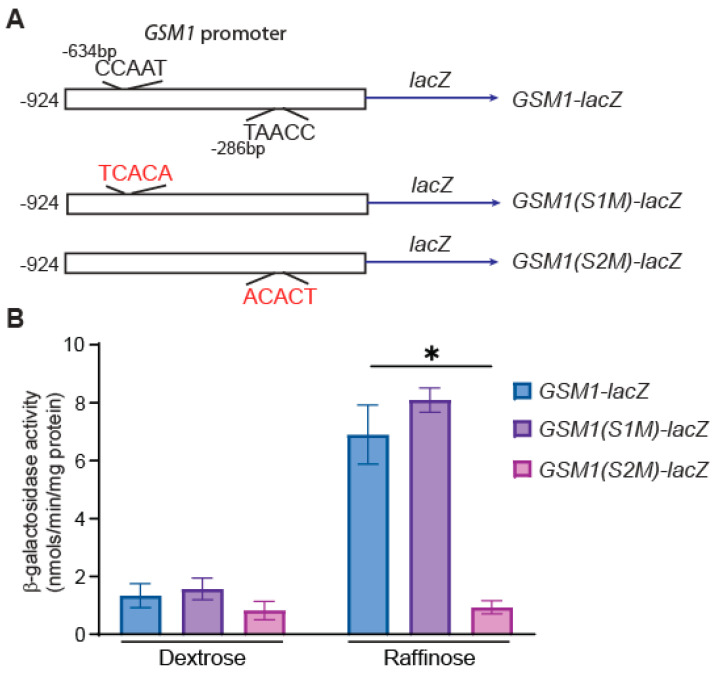
A mutation in the CCAAT sequence at position −286 blocks increased expression of *GSM1* under glucose-derepression conditions. (**A**) Diagrammatic representation of a 924 bp-long promoter sequence of *GSM1* fused to the *lacZ* gene. The *GSM1-lacZ* reporter construct has two CCAAT sequence elements at positions −634 and −286 in relation to the ATG start codon. The mutations to the CCAAT sequences in the *GSM1* promoter are indicated in red. (**B**) Wild-type cells (BY4741) carrying a plasmid encoding *GSM1-lacZ* (pZL3454), *GSM1(S1M)-lacZ* (pMB165)*,* or *GSM1(S2M)-lacZ* (pMB168) were grown in YNBcas5D (dextrose) and YNBcasR (raffinose) medium, and β-galactosidase activity assays were conducted as described in the Materials and Methods. The data are presented as the mean ± standard deviation. ∗, *p* < 0.05.

**Figure 3 genes-15-01128-f003:**
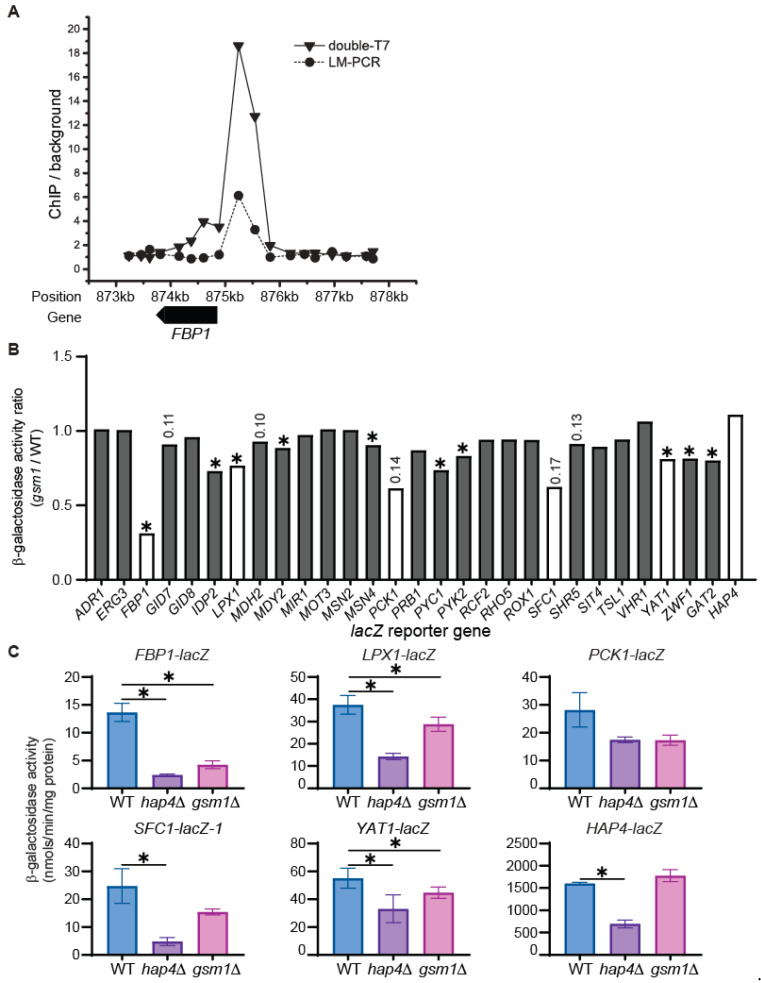
Transcriptional analysis of potential Gsm1 target genes using *lacZ* reporter gene analysis. (**A**) Gsm1-binding ratios of the *FBP1* locus on the chromosome, as determined using double-T7 and LM-PCR methods [31]. (**B**) The expression ratios of 29 *lacZ* reporter genes in *gsm1*Δ mutant cells versus wild-type cells grown in YNBcasR medium. ∗, *p* < 0.05. The numbers on top of the bars are *p* values close to the 0.05 cut-off. The white bars indicate genes selected for further analysis. (**C**) A β-galactosidase activity assay on the expression of *FBP1-, LPX1-*, *PCK1-, SFC1, YAT1-*, and *HAP4-lacZ* reporter genes in wild-type (BY4741), *hap4*Δ (ZLY2811), and *gsm1*Δ (MBY123) mutant cells grown in YNBcasR medium. ∗, *p* < 0.05. *FBP1-lacZ*, pMB179; *LPX1-lacZ*, pMB181; *PCK1-lacZ*, pZL3628; *SFC1-lacZ*, pMB209; *YAT1-lacZ*, pZL3625; *HAP4-lacZ*, pDC124.

**Figure 4 genes-15-01128-f004:**
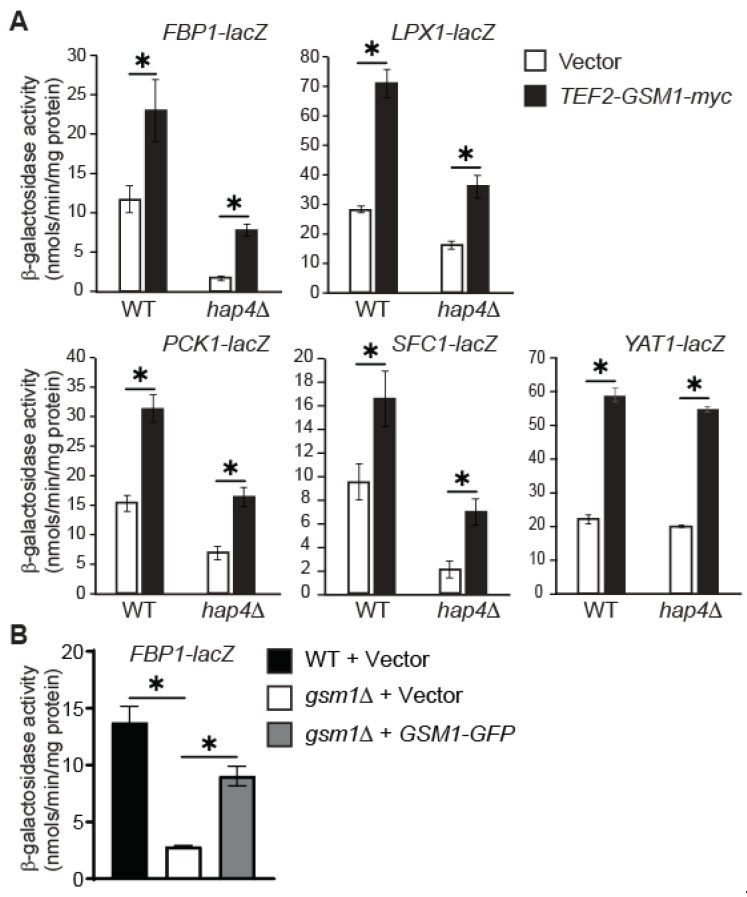
(**A**) *GSM1* overexpression increases the expression of its candidate target genes in *hap4*∆ mutant cells. Wild-type (BY4741) and *hap4*∆ mutant cells (ZLY2811) expressing a *lacZ* reporter gene as indicated were transformed with a centromeric plasmid overexpressing *GSM1* under the control of the *TEF2* promoter (*TEF2-GSM1-myc*, pZL3459) or with the empty vector (Vector, pRS415TEF), which served as a control. Transformants were grown to mid-logarithmic phase in a complete supplement mixture medium with raffinose as the carbon source (CSM-raffinose), and β-galactosidase activity assays were conducted. ∗, *p* < 0.05. (**B**) The *GSM1-GFP* construct was largely functional. Wild-type (BY4741) and *gsm1*∆ mutant cells (MBY123) carrying a plasmid encoding *FBP1-lacZ* and a plasmid encoding *GSM1-GFP* (pZL3613) or carrying the empty vector (Vector, pRS415) were grown to mid-logarithmic phase in CSM-raffinose medium, and β-galactosidase activity assays were conducted. ∗, *p* < 0.05.

**Figure 5 genes-15-01128-f005:**
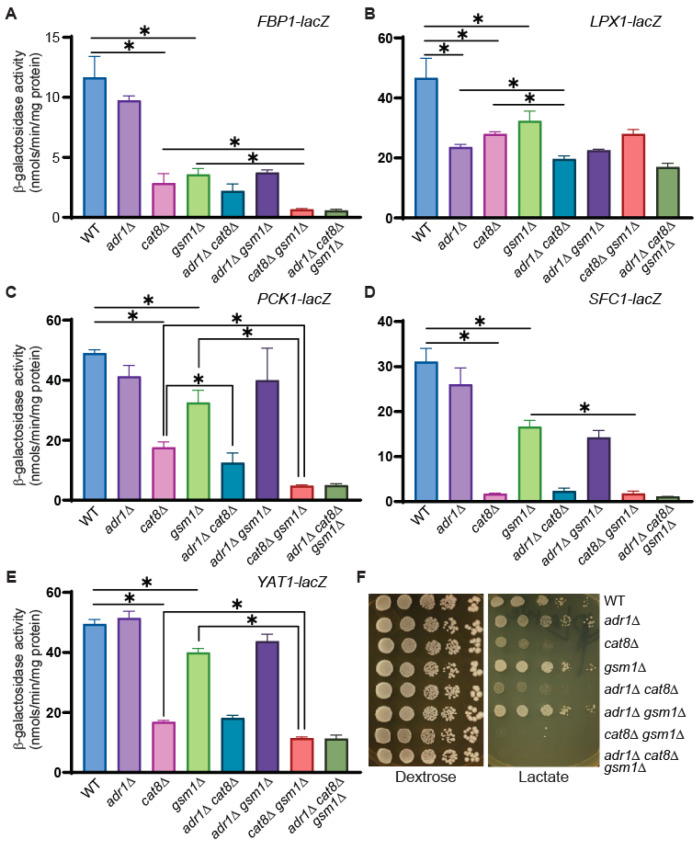
Cat8 and Gsm1 are important in the transcriptional regulation of *FBP1*, *PCK1*, *SFC1*, and *YAT1* and in the utilization of lactate. (**A**–**E**) β-galactosidase activity assays of the expression of *lacZ* reporter genes, as indicated in wild-type (BY4741), *adr1*∆ (ZLY3707), *cat8*∆ (ZLY3701), *gsm1*∆ (MBY123), *adr1*∆ *cat8*∆ (ZLY5048), *adr1*∆ *gsm1*∆ (ZLY5103), *cat8*∆ *gsm1*∆ (ZLY5081)*,* and *adr1*∆ *gsm1*∆ *cat8*∆ (ZLY5109) mutant cells grown in YNBcasR medium. The data are presented as the mean ± standard deviation. ∗, *p* < 0.05. (**F**) *gsm1*∆ exacerbates the growth defect of *cat8*∆ mutant cells grown on lactate medium. Yeast strains as described for panels (**A**–**E**) were serially diluted and spotted on YPD (dextrose) and YPL (lactate) medium. Pictures of the plates were taken after 2–4 days’ growth at 30 °C.

**Figure 6 genes-15-01128-f006:**
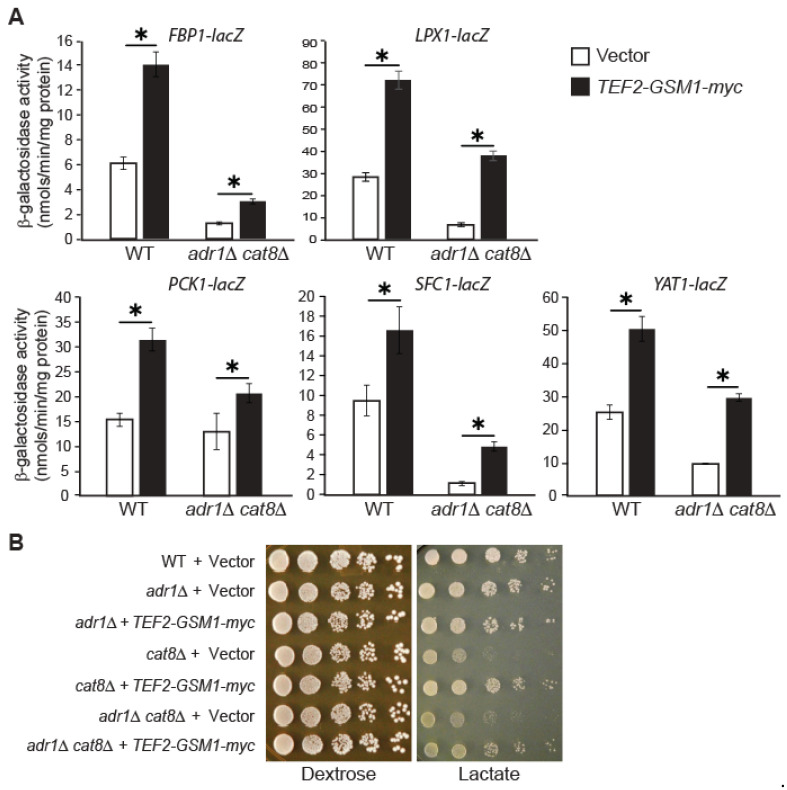
(**A**) *GSM1* overexpression increases the expression of its target genes in *adr1*∆ *cat8*∆ mutant cells. Wild-type (BY4741) and *adr1*∆ *cat8*∆ mutant cells (ZLY5048) expressing the *lacZ* reporter gene, as indicated, were transformed with a centromeric plasmid overexpressing *GSM1* under the control of the *TEF2* promoter (*TEF2-GSM1-myc*, pZL3459) or with an empty vector (Vector, pRS415TEF) as the control. Transformants were grown to mid-logarithmic phase in complete supplement mixture medium (CSM) with raffinose as the carbon source and β-galactosidase activity assays were conducted. ∗, *p* < 0.05. (**B**) *GSM1* overexpression suppresses the growth defects of *cat8*∆ single and *adr1*∆ *cat8*∆ double mutant cells on lactate medium. Wild-type (BY4741), *adr1*∆ (ZLY3707), *cat8*∆ (ZLY3701)*,* and *adr1*∆ *cat8*∆ (ZLY5048) mutant cells carrying a plasmid encoding *TEF2-GSM1-myc* (pZL3459) or carrying the empty plasmid (Vector, pRS415TEF) were serially diluted and spotted on YPD (dextrose) and YPL (lactate) medium. Pictures of the plates were taken after 2–4 days’ growth at 30 °C.

**Table 1 genes-15-01128-t001:** Yeast strains used in this study.

Strain	Genotype	Source
BY4741 (WT)	*MATa ura3 leu2 his3 met15*	Lab stock
ZLY2811 (*hap4*)	BY4741 *hap4::kanMX4*	[34]
MBY123 (*gsm1*)	BY4741 *gsm1::kanMX4*	This study
ZLY3707 (*adr1*)	BY4741 *adr1::kanMX4*	[34]
ZLY3701 (*cat*∆)	BY4741 *cat8::kanMX4*	[34]
ZLY5048 (*adr1 cat8*)	BY4741 *adr1::kanMX4 cat8::HIS3*	This study
ZLY5103 (*adr1 gsm1*)	*MATa ura3 leu2 his3 lys2 adr1::kanMX4 gsm1::kanMX4*	This study
ZLY5081 (*cat8 gsm1*)	BY4741 *gsm1::kanMX4 cat8::HIS3*	This study
ZLY5109 (*adr1 cat8 gsm1*)	*MATa ura3 leu2 his3 lys2 adr1::kanMX4 gsm1::kanMX4 cat8::HIS3*	This study

**Table 2 genes-15-01128-t002:** Plasmids used in this study.

Plasmid	Description	Source
pZL3454	pGSM1-lacZ expressing *lacZ* under the control of a 924 bp *GSM1* promoter in the centromeric plasmid WEJ derived from pRS416 [35].	This study
pZL3462	pRS416-GSM1-GFP expressing Gsm1 from its own promoter with a GFP tag at the C-terminus.	This study
pMB165	pGSM1(S1M)-lacZ expressing *lacZ* under the control of a *GSM1* promoter with a mutation in the CCAAT sequence at position –634 in relation to the ATG start codon.	This study
pMB168	pGSM1(S2M)-lacZ expressing *lacZ* under the control of a *GSM1* promoter with a mutation in the CCAAT sequence at position −286 in relation to the ATG start codon.	This study
pMB179	pFBP1-lacZ expressing *lacZ* under the control of a 924 bp *FBP1* promoter.	This study
pZL3628	pPCK1-lacZ expressing *lacZ* under the control of a 1484 bp *PCK1* promoter.	This study
pMB209	pSFC1-lacZ expressing *lacZ* under the control of a 1454 bp *SFC1* promoter.	This study
pMB181	pLPX1-lacZ expressing *lacZ* under the control of a 1039 bp *LPX1* promoter.	This study
pZL3625	pYAT1-lacZ expressing *lacZ* under the control of a 1555 bp *YAT1* promoter.	This study
pDC124	pHAP4-lacZ expressing *lacZ* under the control of a 1.8 kbp *HAP4* promoter.	[36]
pZL3613	pRS415-GSM1-GFP expressing Gsm1 from its own promoter with a GFP tag at the C-terminus.	This study
pZL3459	pRS415-TEF2-GSM1-myc, expressing Gsm1 with a C-terminal 3x myc epitope tag under the control of the strong *TEF2* promoter. The plasmid was constructed from pRS415TEF [37].	This study
pRS415TEF	pRS415-TEF2p-CYC1 terminator.	[37]

## Data Availability

The original contributions presented in the study are included in the article, further inquiries can be directed to the corresponding author.

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
