# Peer review of "GSM1 Requires Hap4 for Expression and Plays a Role in Gluconeogenesis and Utilization of Nonfermentable Carbon Sources"

_genes, 2024, doi:10.3390/genes15091128_

Round 1
Reviewer 1 Report
Comments and Suggestions for Authors
Title: Gsm1 Requires Hap4 for Expression and Plays a Role in Gluconeogenesis and Utilization of Nonfermentable Carbon Sources
A reviewer suggested comments mentioned below:
The manuscript describes the role of gsm1 and Hap4 in glucose metabolism at glucose depression conditions in Yeast as a carbon source.
The abstract be shorten and more crisp to describe your study. Abstract description should be focus on your worked what you have done including background, methodology and results found. Authors need to revise thoroughly to revise each part of the section carefully.
Methodology must be revised, according to the mentioned results section, methods lack an accurate description of the results. SDS PAGE run gel picture clarity did not show up to mark for publication. Anti pgk1 gel running band should be revised and a good quality band picture mentioned in the manuscript. Only Figure 1 has mentioned gel picture but lacks in other mentioned figures where authors showed gel picture. Should be mentioned all the run gel picture in a results as a representative picture in the manuscript.
The authors showed only protein-level knockout and gene-level expression of the mentioned genes in the manuscript. In method, section should also describe gene-level expression methodology.
The concentration or dilution of antibodies used in the results should be mentioned in the methods.
Conclusion and limitation of the resulted data can be mentioned by the authors in the revised manuscript.
A few places need to be changed and revised in the sentences:
Line 176-77: Should be rechecked the sentence mentioned: “Similar results were also obtained from an otherwise wild-type strain expressing GSM1- 176 GFP (Fig. 1C).”
Comments on the Quality of English Language
English language is satisfactory but need to revise the manuscript once thoroughly.
Author Response:
We thank the reviewer for a careful reading of our manuscript and constructive comments. We have revised our manuscript based on these comments. Our point-by-point responses are as follows.
The abstract be shorten and more crisp to describe your study. Abstract description should be focus on your worked what you have done including background, methodology and results found. Authors need to revise thoroughly to revise each part of the section carefully.
We have shortened the abstract as suggested by the reviewer.
Methodology must be revised, according to the mentioned results section, methods lack an accurate description of the results. SDS PAGE run gel pict equivalent amounts ofure clarity did not show up to mark for publication. Anti pgk1 gel running band should be revised and a good quality band picture mentioned in the manuscript. Only Figure 1 has mentioned gel picture but lacks in other mentioned figures where authors showed gel picture. Should be mentioned all the run gel picture in a results as a representative picture in the manuscript.
We made significant changes to the “Cellular extract preparation, immunoblotting, and immunoprecipitation” section in the Materials and Methods (lines 123-138).
Gsm1-GFP blot quality: The Gsm1-GFP signal was very weak. We are reluctant to push the contrast and enhance image to the point of being artificial.
Anti-Pgk1 band: The SDS-PAGE gel lanes were loaded equivalent amounts of cellular proteins based on OD600 reading of cell cultures (lines 131-132 in the revised manuscript). Pgk1 was included as a secondary, visual loading control. Since Gsm1-GFP shows an all-or-none expression on western blots, a precise quantification of anti-Pgk1 bands is unnecessary.
Only Figure 1 has mentioned gel picture but lacks in other mentioned figures where authors showed gel picture: Only Figure 1 has gel pictures. The others do not. We are not sure about the question being asked here.
Representative gel pictures: We have added the statement, “The result was representative of three independent experiments for panel (B) and two for panel (C),” at the end of the Figure 1 legend (lines 194-195).
The authors showed only protein-level knockout and gene-level expression of the mentioned genes in the manuscript. In method, section should also describe gene-level expression methodology.
Gene-level expression was analyzed using a beta-galactosidase activity assay, which was described in the manuscript submitted for review.
The concentration or dilution of antibodies used in the results should be mentioned in the methods.
The dilution of the antibodies was now included in the method (lines 135-138).
Conclusion and limitation of the resulted data can be mentioned by the authors in the revised manuscript.
One limitation of this study is the dependence of lacZ reporter genes as a readout of gene expression. But it’s also the strength because it would be difficult for RT-PCR and Northern blotting to pick up a difference of less than 30%. In Figure 3B, most reporter genes have a difference of less than 30% in wild type versus the gsm1D mutant.
Line 176-77: Should be rechecked the sentence mentioned: “Similar results were also obtained from an otherwise wild-type strain expressing GSM1- 176 GFP (Fig. 1C).”
In Figure 1B, GSM1-GFP expression was examined in gsm1D mutant cells. In Figure 1C, GSM1-GFP expression was examined in wild type and isogenic hap4D mutant cells. The statement correctly described what was done and the result.
Reviewer 2 Report
Comments and Suggestions for Authors
The manuscript shows results of the study of GSM1 gene expression and Gsm1 protein function as transcriptional activator important in gluconeogenesis and growth on nonfermentable carbon sources of Saccharomyces cerevisiae cells. The study is systematic and comprehensive, adds to the better understanding of complex transcriptional regulation of gluconeogenesis genes and complements nicely the recent study from the other lab.
General Comments
The manuscript is clearly and well written. However, small mistakes require correction.
Genes are expressed, not proteins. This was correctly stated throughout the manuscript but not in the title. The title rather should be “GSM1 requires Hap4 for expression…. “
Mutant names (relevant genotypes) are not consistent in figures. In figures 3C, ,4B, 5 and 6 delta (Δ) is omitted but in 4A is present. Better unify, give correct mutant gene names in full.
Specific comments
Page 1, L69 GSM1/YJL103C Systematic name should be given for GSM1 in Introduction as it first appears. On Page 4, L157 is too late.
Page 3, Table 2, row 5 pMB179 should be : ….FBP1 promoter
row 12 should be: pRS415-TEF2-GSM1-myc to be consistent with the nomenclature in other rows of table 2 and with the text.
Page 4, L116, 117 “*” in superscript is hardly readable. Last sentence of the paragraph could be omitted in the Materials and Methods, this information in the legends of figures is sufficient.
L157 double (systematic) name should be given earlier in the introduction. Why here?
Page 6, Figure 4 legend , L186 “*” is hard to read, better just: *, p <0.05
Page 7, L205 In contrast, the
Page 8, L240 expression “construct” is a laboratory jargon. Plasmids bearing lacZ reporter genes were transformed…
Page 9 , Figure 3C Haw many repetitions? Show statistical analysis of results, significance.
Page 10, L295, 302, should be TEF2-GSM1-myc, to be consistent with the Table 2, nomenclature used and the text.
Page 11, Figure 4A and the legend L317, should be TEF2-GSM1-myc.
Figure 4B Vector, not Vecotr
Page 12, L348, better add A or The, do not start a sentence with small letter.
Page 13 Figure 5, How many repeats? Show statistical analysis, significance of results.
L374 do not split 30OC
Page 15, Figure 6, make the figure and legend consistent with the Table 2 and text. Should be TEF2-GSM1-myc in A and B. Make the image 4B smaller, as is in 5F. L409, do not have to repeat so many times the same informtion, is obvious.
Author Response:
We thank the reviewer for a careful reading of our manuscript and constructive comments. We have revised our manuscript based on these comments. Our point-by-point responses are as follows.
Genes are expressed, not proteins. This was correctly stated throughout the manuscript but not in the title. The title rather should be “GSM1 requires Hap4 for expression…. “
Revised as suggested.
Mutant names (relevant genotypes) are not consistent in figures. In figures 3C, ,4B, 5 and 6 delta (Δ) is omitted but in 4A is present. Better unify, give correct mutant gene names in full.
We have added “D” to all mutant gene names in the figures.
Specific comments
Page 1, L69 GSM1/YJL103C Systematic name should be given for GSM1 in Introduction as it first appears. On Page 4, L157 is too late.
Revised as suggested.
Page 3, Table 2, row 5 pMB179 should be : ….FBP1 promoter
Revised as suggested.
row 12 should be: pRS415-TEF2-GSM1-myc to be consistent with the nomenclature in other rows of table 2 and with the text.
Revised as suggested.
Page 4, L116, 117 “*” in superscript is hardly readable. Last sentence of the paragraph could be omitted in the Materials and Methods, this information in the legends of figures is sufficient.
We changed “*” font type to symbol throughout the manuscript and deleted the last sentence of the paragraph as suggested.
L157 double (systematic) name should be given earlier in the introduction. Why here?
We deleted YJL103C.
Page 6, Figure 4 legend, L186 “*” is hard to read, better just: *, p <0.05.
We changed “*” Palatino Linotype font to symbol for easy reading in all figure legends. We also updated “*” to “*” in Figure 3B accordingly. We also adopted “*, p <0.05” in most figure legends.
Page 7, L205 In contrast, the
Revised as suggested.
Page 8, L240 expression “construct” is a laboratory jargon. Plasmids bearing lacZ reporter genes were transformed…
Revised as suggested (line 241)
Page 9 , Figure 3C Haw many repetitions? Show statistical analysis of results, significance.
The Figure 3C data on the lacZ reporter genes in wild type versus gsm1D were the same as presented in Figure 3B. The information on the repetitions was described on lines 444-446 in the original manuscript. A statistical analysis of the results (p < 0.05) is now included in Figure 3C, and the Figure 3C legend was revised accordingly.
Page 10, L295, 302, should be TEF2-GSM1-myc, to be consistent with the Table 2, nomenclature used and the text.
We changed TEF-GSM1-myc to TEF2-GSM1-myc throughout the manuscript.
Page 11, Figure 4A and the legend L317, should be TEF2-GSM1-myc.
Revised as suggested.
Figure 4B Vector, not Vecotr
Corrected the typo as suggested.
Page 12, L348, better add A or The, do not start a sentence with small letter.
Revised as suggested (line 345).
Page 13 Figure 5, How many repeats? Show statistical analysis, significance of results.
There were four repeats for most strain/plasmid combinations. Some had fewer than four due to contamination during overnight culturing of cells. Because of the large number of possible pairwise comparisons in each figure panel, only the results from statistical analysis of selective comparisons with p < 0.05 are now included in Figure 5.
L374 do not split 30OC
Revised as suggested.
Page 15, Figure 6, make the figure and legend consistent with the Table 2 and text. Should be TEF2-GSM1-myc in A and B. Make the image 4B smaller, as is in 5F. L409, do not have to repeat so many times the same information, is obvious.
TEF2-GSM1-myc is now used in the figure and figure legends. The size of image 4B was reduced. “*, p < 0.05” is used in all figure legends except for the first time use in Figure 1.
The authors’ finding:
The mutations in Figure 2A were not colored red. This has been corrected.
Round 2
Reviewer 1 Report
Comments and Suggestions for Authors
The revised version of the manuscript has been updated by the authors with all the suggested comments by the reviewer. Reference must be rechecked that mentioned references between texts are appropriate according to written information.
Author Response
Response to reviewer #1’s comment (round 2)
We have checked the references one by one. We removed one of the three references on online 68 (viewing with No Markup) and corrected a reference on line 110. Thank you.
Zhengchang Liu